# Effect of Ionic Liquids in the Elaboration of Nanofibers of Cellulose Bagasse from *Agave tequilana* Weber var. *azul* by Electrospinning Technique

**DOI:** 10.3390/nano12162819

**Published:** 2022-08-17

**Authors:** Enrique Márquez-Ríos, Miguel Ángel Robles-García, Francisco Rodríguez-Félix, José Antonio Aguilar-López, Francisco Javier Reynoso-Marín, José Agustín Tapia-Hernández, Francisco Javier Cinco-Moroyoqui, Israel Ceja-Andrade, Ricardo Iván González-Vega, Arturo Barrera-Rodríguez, Jacobo Aguilar-Martínez, Edgar Omar-Rueda-Puente, Carmen Lizette Del-Toro-Sánchez

**Affiliations:** 1Departamento de Investigación y Posgrado en Alimentos, Universidad de Sonora, Blvd. Luis Encinas y Rosales S/N, Hermosillo 83000, Sonora, Mexico; 2Departamento de Ciencias Médicas y de la Vida, Centro Universitario de la Ciénega, Universidad de Gaudalajara, Av. Universidad 1115, Ocotlán 47820, Jalisco, Mexico; 3Departamento de Genómica Alimentaria, Universidad de la Ciénega del Estado de Michoacán de Ocampo (UCEMICH), Avenida Universidad 3000, Colonia Lomas de la Universidad, Sahuayo 59103, Michoacan, Mexico; 4Departamento de Física, Centro Universitario de Ciencias Exactas e Ingenierías, Universidad de Guadalajara, Blvd. M. García-Barragán 1451, Guadalajara 44430, Jalisco, Mexico; 5Departamento de Ciencias Básicas, Centro Universitario de la Ciénega, Universidad de Gaudalajara, Av. Universidad 1115, Ocotlán 47820, Jalisco, Mexico; 6Departemento de Ciencias Tecnológicas, Centro Universitario de la Ciénega, Universidad de Gaudalajara, Av. Universidad 1115, Ocotlán 47820, Jalisco, Mexico; 7Departamento de Agricultura y Ganadería, Universidad de Sonora, Blvd. Luis Encinas y Rosales S/N, Hermosillo 83000, Sonora, Mexico

**Keywords:** agave bagasse cellulose, cellulose nanofibers, electrospinning technique, ionic liquids

## Abstract

The objective of this paper was to report the effect of ionic liquids (ILs) in the elaboration of nanofibers of cellulose bagasse from *Agave tequilana* Weber var. *azul* by the electrospinning method. The ILs used were 1-butyl-3-methylimidazolium chloride (BMIMCl), and DMSO was added as co-solvent. To observe the effect of ILs, this solvent was compared with the organic solvent TriFluorAcetic acid (TFA). The nanofibers were characterized by transmission electron microscopy (TEM), X-ray, Fourier transform-infrared using attenuated total reflection (FTIR-ATR) spectroscopy, and thermogravimetric analysis (TGA). TEM showed different diameters (ranging from 35 to 76 nm) of cellulose nanofibers with ILs (CN ILs). According to X-ray diffraction, a notable decrease of the crystalline structure of cellulose treated with ILs was observed, while FTIR-ATR showed two bands that exhibit the physical interaction between cellulose nanofibers and ILs. TGA revealed that CN ILs exhibit enhanced thermal properties due to low or null cellulose crystallinity. CN ILs showed better characteristics in all analyses than nanofibers elaborated with TFA organic solvent. Therefore, CN ILs provide new alternatives for cellulose bagasse. Due to their small particle size, CN ILs could have several applications, including in food, pharmaceutical, textile, and material areas, among others.

## 1. Introduction

*Agave tequilana* Weber var. *azul* bagasse is an agro-industrial waste of the tequila industry [1,2]. According to the Tequila Regulatory Council (Consejo Regulador del Tequila), in the year 2021, the consumption of agave was 2018.7 thousand tons [3], and Cedeño [4] indicated that the bagasse production is equivalent to 40% of the weight of the head agave. This quantity corresponds to 807.48 thousand tons of agave bagasse produced. This agro-industrial waste contains principally cellulose (42%) [5]. This compound has gained importance in several applications (reinforcing materials, packaging, and fabricating composite materials, among others), especially at nanometer scales [6,7]. Currently, society demands green, non-toxic, and sustainable materials [8]. Hence, obtaining cellulose from this waste could be a good option and could be used to produce nanofibers from a bioeconomic point of view.

However, although cellulose is the most abundant renewable and biodegradable polymer, its structure consists of repeated D-glucose units which form strong intermolecular and intramolecular hydrogen bonds that make it insoluble in water and most organic solvents [9,10,11,12,13]. For this reason, in recent years, ionic liquids (ILs) have attracted much attention as a solvent for cellulose by disrupting hydrogen bonding with applied heat. ILs are salts or pairs of ions having a melting temperature at or close to ambient temperature (≤100 °C) and sometimes even below room temperature, as well as high electrochemical and thermal stability, and their main properties can be tailored by the proper selection of anions and cations [9,14,15,16,17,18,19,20]. ILs are nonvolatile solvents due to their significantly high vapor enthalpies; they consist of numerous different cation and anion combinations which influence their physicochemical properties. ILs are novel green solvents typically consisting of organic cations and organic or inorganic anions based on imidazolium, ammonium, and phosphonium, combined to form an ionic liquid. Furthermore, ILs enable the solvent to be designed and tuned to optimize yield, selectivity, and solubility [21].

ILs have particular advantages such as tunable structure, excellent dissolution ability, high thermal stability, non-flammability, low vapor pressure, high ionic conductivity, miscibility with many other solvent systems, and ability to be separated and recovered easily in comparison with conventional solvents [14,17,22,23,24,25,26,27,28]. In recent years, some studies have focused on using ILs as a solvent to dissolve cellulose, such as 1-butyl-3-methylimidazolium chloride (BMIMCl) and 1-allyl-3-methylimidazolium chloride (AMIMCl), which can dissolve cellulose without derivation. The good dissolving capabilities of these ILs have been directly correlated with their strong hydrogen bond basicity and lower viscosity. Furthermore, the chemical structure of imidazolium cations also has an important influence on cellulose dissolution: when the chain length of alkyl groups or the symmetry of cations increased, the dissolution rate of cellulose in ILs decreased because of the increase in viscosity or/and the decrease in hydrogen bond acidity [24].

The principal role of the high chloride concentration in BMIMCl is breaking the hydrogen bonding of cellulose. The anions and cations in ILs play a predominant role by the formation of strong hydrogen bonds with hydrogen atoms of hydroxyls in cellulose. The relatively small anions favor the formation of hydrogen bonds with hydrogen atoms of hydroxyls. The aromatic protons in the imidazolium cation, especially H_2_, prefer to associate with oxygen atoms of hydroxyls with less steric hindrance [24]. Another characteristic of this solvent is its high stability at ambient pressure, and it is also the greatest dissolver of up to 10 wt% concentration of cellulose by heating; furthermore, such solvents’ biocompatibility and biodegradability make them suitable for biomedical applications [14,22,26,27]. Furthermore, the addition of polar and aprotic co-solvents such as dimethyl sulfoxide (DMSO) would accelerate the dissolution of cellulose, modifying the surface tension and electrical conductivity. The addition of co-solvents produced thinner and more uniform fibers. In addition, DMSO would enhance their solvating power and accelerate mass transport by decreasing the solvent viscosity without having a significant influence on the specific interactions between cations and anions or between ILs and cellulose [24,26]. Therefore, cellulose solutions in BMIMCl can be easily precipitated by only adding water, ethanol, or acetone, and separating the solvent after cellulose precipitation, [14], as the ILs are nearly completely recovered (<99.5%) after use in the process and then can be reused [28].

Recently, electrohydrodynamic atomization (EHDA) techniques (electrospinning, electrospraying, and e-jet printing) have become part of a boom in obtaining micro- and nanomaterials [29,30]. Specifically, the electrospinning technique has received growing interest due to it being a novel method that uses the high electrostatic force applied to a polymer solution or melt to elaborate oriented polymer fibers; it allows fine control of the fibers’ properties and can tune them to the individual needs of specific applications, with diameters in the micro- to nanometric scale [20,23,31,32,33,34]. This technique uses a high-voltage power to apply an electric field to solutions between the needle tip and two electrodes bearing electrical charges of opposite polarity, one placed onto the spinneret and the other onto the collector. The charged solution jet evaporates during its travel towards the collector to form the nonwoven fibers’ arrangement. The potential difference may be heightened to produce the desirable morphology, depending on solution properties such as polymer molecular weight, conductivity, surface tension, and viscosity [23,26]. Electrospinning can produce nanofibers from a large variety of polymers, both natural and synthetic [35,36]. In this method, it is essential to find the appropriate condition to form a stable conical droplet known as a Taylor cone to produce continuous fibers. It is important to consider some solution parameters that can affect the electrospinning process: concentration, viscosity, surface tension, conductivity, dielectric constant, solvent volatility, and the polymer molecular weight [35]. In the process, some other parameters are important to consider: applied voltage, electrostatic potential, electric field strength, electrostatic field shape, working distance, and solution flow rate. It is necessary to consider the temperature, humidity, atmospheric composition, and pressure as environmental conditions. These factors can seriously affect this technique because they may affect the morphology and properties of the particles [9].

Nanofibers elaborated by electrospinning have high flexibility, ordered morphology, specific functionalities, high porosity, small pore size, high surface area, and excellent mechanical properties [26,34,37]. These kinds of nanoparticles can be applied in different fields such as agriculture, pharmacy, food, cosmetics, textiles, biomedicine, scaffolding for tissue engineering, protein purification, protective clothing, electronic and optical devices, pesticide spraying, and enzyme and catalytic supports [23,35,38,39,40].

Therefore, the objective of this research was to report the fabrication and the evaluation of some physicochemical properties of cellulose nanofibers from agave bagasse prepared by the electrospinning technique using ILs as solvent.

## 2. Materials and Methods

### 2.1. Reagents and Materials

Reagents such as 1-butyl-3-methylimidazolium chloride (BMIMCl), trifluoracetic acid (TFA) 99%, and dimethyl sulfoxide (DMSO) were from Sigma-Aldrich (St. Louis, MO, USA). The rest of the chemical reagents were of analytical grade.

*Agave tequilana* Weber var. *azul* bagasse was donated by a tequila-producing company located in Jalisco, Mexico (geographic coordinates: −102.549278 W, 20.528508 N) at 1761 m above sea level.

### 2.2. Extraction of Cellulose

Cellulose was obtained according to the methodology proposed by Robles-García and coworkers [41]. Briefly, agave bagasse was dried using a tunnel dryer at 80 °C for 2 h. The dried material was milled in a hammer mill at 3000 rpm and sieved to a final particle size of 190 μm. Following that, 70 g of milled agave bagasse were mixed with 1 L of 1.049 M CH₃COOH containing 0.5 M NaOH. Digestion was carried out for 210 min at 70 °C after adding 400 mL of 20% (*w*/*v*) NaClO. The holocellulose content (α–cellulose+hemicelluloses) was then obtained with NaClO_2_ (20% *w*/*v*) and NaOH (0.5 M). α–Cellulose was separated from hemicelluloses with 17.5% (*w*/*v*) NaOH and dried at 105 °C for 24 h.

### 2.3. Preparation of Polymer Solution Using Ionic Liquids (ILs) Solvent

First, 1.5 g of BMIMCl was heated at 90 °C in an oil bath. When melted completely, 0.015 g of agave bagasse cellulose and 0.001 g of DMSO were added as co-solvents. This mixture was vigorously stirred for 1 h. Following that, 10 mL of polymer solution was transferred into a glass syringe with a needle of 0.8 mm diameter. The glass syringe was covered with a silicone rubber heating tape with adjustable thermostat control (HSTAT), and electrospinning was carried out, as indicated later. To find conditions for nanofiber formation, we prepared six different polymer solutions, as shown in Table 1. DMSO was added to solutions with the intent to improve the swelling of cellulose in BMIMCl solvent to reduce the viscosity and facilitate the electrospinning process.

### 2.4. Nanofiber Elaboration by Electrospinning Process

The equipment consisted of three parts: a high-power source with a voltage of 17 kV, a silicone rubber heating tape with adjustable thermostat control (HSTAT) that covers the syringe, and a water bath collector (Figure 1). The glass syringe was placed vertically above the water bath collector to remove ionic liquid from the nanofibers. The distance between the needle tip and the water bath was 15 cm. The flow rate of the polymeric solution depended only on gravity. The fibers formed were immersed in 100 mL deionized water at 70 °C for 45 min to remove BMIMCl (three washes) and dried in a vacuum oven at 60 °C for 24 h [42], which corresponded the cellulose nanofibers with ionic liquids (CN ILs). To observe the effect of ILs against organic solvents, CN ILs were compared with cellulose nanofibers prepared with the organic solvent (CN) trifluoracetic acid as indicated by Robles-García and coworkers [41].

### 2.5. Characterization of Cellulose Nanofibers Elaborated with IL

#### 2.5.1. Transmission Electron Microscopy (TEM)

TEM images of electrospun fibers were performed using a JEOL JEM-1010 electron microscope (JEOL, Tokyo, Japan) operated at 80 kV. Drops (about 10 μL) of cellulose nanofiber suspension were deposited onto a carbon film-coated copper grid (200 mesh). The grid was allowed to dry at 25 °C for 2 days and then stored in a desiccator until use.

#### 2.5.2. X-ray Diffraction (XRD)

The nanofibers were analyzed by X-ray diffraction (XRD) using a theta–theta model X-ray diffractometer (from STOE-Germany) with 35 kV power and 30 mA of current. The X-ray diffraction patterns were recorded at 25 °C in the 2*θ* angle range between 5° and 80°, with a 2*θ* step of a scanner of 0.02°, for 60 s per point and using Cu Kα (λ = 0.154 nm) radiation.

#### 2.5.3. Fourier Transform Infrared with Attenuated Total Reflectance (FTIR-ATR)

To determine the composition and structural changes in the agave bagasse, agave cellulose, CN, and CN ILs produced by electrospinning, samples were analyzed at 25 °C using a Perkin Elmer FT-IR Spectrometer (spectrum Two) FTIR- ATR with a diamond crystal. Spectra were operated in a wavenumber range of 4000–500 cm^−1^. Each measurement was an average of 16 scans per spectrum.

#### 2.5.4. Thermogravimetric Analysis (TGA)

Thermogravimetric analysis was used to study the thermal stability of agave bagasse (AB), agave cellulose (AC), cellulose nanofibers with organic solvents (CN), and cellulose nanofibers with ionic liquids (CN ILs). TGA was performed in a Perkin Elmer TGA analyzer, and the thermograms were obtained from the Pyris 1 software. The mass of each dried sample was 10 mg. The analysis was carried out from 25 to 900 °C, and two different atmospheres were used. The first experiment was under nitrogen flow at 25 mL min^−1^ using a constant heating rate of 10 °C min^−1^. The second one was under airflow at 25 mL min^−1^ and the same constant heating rate from 600 to 900 °C. The experiments were performed in triplicate.

## 3. Results and Discussion

### 3.1. Elaboration Conditions of CN ILs

Several polymer concentrations and different electrospinning conditions were tested to obtain CN ILs. The best conditions to elaborate these nanofibers were: polymer concentration of 4%, a system of ILs solvent with BMIMCl 15% (*v*/*v*), a flow rate dependent only on gravity, and a voltage applied of 17 kV. The syringe and the HSTAT silicone rubber heating tape were placed vertically above the collector at a distance of 15 cm from the needle tip and the water bath. There are reports about the application and advantages of some nanoparticles elaborated with ILs. For example, the group of Xu [11] prepared cellulose-*graft*-polycaprolactone (*g*-PCL) copolymers using ILs as a solvent and used them to adsorb Cu(II) and Ni(II) ions from aqueous solutions. They reported that cellulose-*g*-PCL exhibited a satisfactory removal capacity of 98.7% for Cu(II) and 35.6% for Ni(II) ions in aqueous solutions. More recently, Wang and coworkers [43] elaborated cellulose/multi-walled carbon nanotubes (MWCNTs) composite membranes to be applied in electrochemical and biomedical fields using ILs as a solvent. They reported that the binary ionic liquids system could regulate the properties of the composite membranes and efficiently improve their conductivity, mechanical properties, and thermal stability by promoting the dispersion of MWCNTs. Ninomiya and coworkers [44] used lignocellulose nanofibers prepared by pretreatment with ILs as nanocomposites with polypropylene to produce so-called wood–plastic composites (WPCs) or natural fiber–plastic composites.

Cellulose fibers have been extensively studied over the last two decades. However, this study focuses on the production of cellulose nanofibers from agave bagasse using ILs as a solvent and obtained by electrospinning. Therefore, this is an alternative use for this agro-industrial waste that could be employed in many applications in different industrial sectors.

### 3.2. Transmission Electron Microscopy (TEM)

TEM was used to investigate the morphology and diameter of CN ILs compared with the nanofibers with organic solvents (CN) obtained by electrospinning (Figure 2). The nanofiber diameter of CN ILs was from 35 to 76 nm, obtained by analyzing 14 different samples. In comparison, CN measured from 59.34 to 163 nm, approximately. The difference between these diameters could be attributed to the fact that the inclusion of ILs in the polymer matrix induces a decrease in the fiber orientation [20].

The morphology of both samples was a thin fiber without branches and with a regular surface. As the CN ILs become smaller, the surface area is greater, and there is more area for contact with the fiber. Thus, the use of ILs decreases the particle size by half at least compared with CN, indicating that the cellulose could be more available to interact in the formation of nanofiber due to its better solubility caused by the ILs. This advantage could be useful to apply CN ILs to several areas, such as to elaborate mask filters to protect against coronaviruses [45], to elaborate films for enhancing biodegradability in marine environments [46], to support material or encapsulate functional compounds with biological activities or sensors [47], and to fabricate composite hydrogels with potential applications as novel materials in food and non-food applications [48], among others.

Some studies obtained different particle sizes, but this depends on the conditions used. The group of Xu [49] noted that cellulose solution at 5% can produce nanofibers with diameters of 100–300 nm. Furthermore, Quan and coworkers [42] elaborated cellulose nanofibers using BMIMCl as a solvent, obtaining fibers with a minimum diameter of between 500 and 800 nm, much greater than those obtained in our study. In contrast, Ninomiya and colleagues [44] reported a pretreatment of bagasse powder with cholinium ionic liquid for subsequent mechanical nanofibrillation to produce lignocellulose nanofibers with a thickness of approximately 10–20 nm.

### 3.3. X-ray Diffraction (XRD)

Figure 3 shows the XRD patterns of agave bagasse (AB), agave cellulose (AC), cellulose nanofibers with organic solvents (CN), and cellulose nanofibers treated with ionic liquids (CN ILs). The XRD pattern of agave bagasse (AB) exhibits a diffraction peak located at 2θ = 14.9° and a broad peak centered at about 21.8°, which are assigned to the hkl (1 1 0) and hkl (2 0 0) crystalline planes of cellulose, respectively [50,51,52]. Furthermore, the XRD pattern of AB shows diffraction peaks located at 24.4°, 30.1°, and 38.2°, corresponding to monoclinic calcium oxalate monohydrate (CaC_2_O_4_ H_2_O). The presence of calcium oxalate monohydrate in agave bagasse (AB) and agave cellulose (AC) is in good agreement with the reported data for untreated agave bagasse [53,54,55]. Except for a slight shifting of the broad diffraction peak of the hkl (2 0 0) crystalline plane of cellulose to a lesser 2θ angle (≅20.2°) and a slight decrease in the intensity of the diffraction peaks corresponding to the agave bagasse, the XRD pattern of agave cellulose (AC) is very similar to that of agave bagasse (AB). A notable transformation of the crystalline structure of cellulose is observed in the XRD pattern of the cellulose treated with ionic liquids (CN ILs). The sharp diffraction peaks observed in the XRD patterns of AB and AC completely disappeared, and the XRD pattern of CN ILs indicates that the treatment of cellulose with ionic liquids (ILs) influences the loss of crystallinity of cellulose, probably due to the ionic liquid destroying the microcrystalline cellulose fibbers and the calcium oxalate structure. Furthermore, broad diffraction peaks appeared at 2θ angles of about 12°, 20.2°, and 22.2° in the XRD pattern of cellulose treated with ionic liquid (CN ILs). These diffraction peaks correspond to the hkl (1 -1 0), hkl (1 1 0), and hkl (0 2 0) crystalline planes of the polymorph cellulose II-type with a monoclinic P2_1_ unit cell (*a* = 0.810 nm, *b* = 0.903 nm, *c* = 1.031 nm, γ = 117.1°) [56,57]. The polymorph cellulose II-type can be obtained by chemical regeneration or mercerization of natural cellulose [57,58,59]. In our study, the polymorph cellulose II-type was obtained by treating agave cellulose nanofibers with the ionic liquid 1-butyl-3-methylimidazoline, which dissolves the cellulose. Its main role is breaking the hydrogen bonding of cellulose, transforming the agave cellulose structure into the polymorph cellulose II-type structure. On the other hand, the XRD pattern of the cellulose nanofibers is similar to that corresponding to amorphous materials, indicating the low crystallinity of the cellulose nanofibers. Only small diffraction features located at 2θ angles of about 24.4° and 38.2°, corresponding to the remaining calcium oxalate, are observed.

### 3.4. FTIR-ATR Spectroscopy

FTIR was used to analyze spectra of agave bagasse and its delignification to obtain cellulose, as well as the physical interaction between cellulose nanofiber with solvents (organic solvent and BMIMCl). Figure 4 shows spectra of FTIR of agave bagasse (AB), agave cellulose (AC), and cellulose nanofibers prepared by two different solvents, the cellulose nanofibers with organic solvents (CN) and cellulose nanofibers with ionic liquids (CN ILs). All spectra have typical bands: a broad band at 3420 cm^−1^ attributed to the O-H stretching vibration of hydroxyl group of cellulose. This band is broader and shifted to 3332 cm^−1^ by the presence of N-H stretching vibration in the BMIMCl solvent in the spectrum of CN ILs. The band with medium intensity in both spectra at 2980 cm^−1^ shows C-H stretching vibrations by CH and CH_2_ in the chain. The band at 1642 cm^−1^ in all spectra is attributed to adsorbed water in cellulose [60,61]. The agave bagasse spectrum has two medium characteristic bands due to the presence of lignin, 1513 cm^−1^ by C=C stretching in the aromatic ring, and 1235 cm^−1^ by C-O stretching of phenolic hydroxyl groups. These bands decrease significantly for cellulose and cellulose nanofibers due to the delignification process [55]. A narrow and strong band at 1019 cm^−1^ is related to the C-O-C vibrations of the pyranose ring of cellulose and both cellulose nanofibers [62]. However, these vibrations are displaced at 1052 cm^−1^ in the agave bagasse spectrum. The band at 1142 cm^−1^ is due to the CO symmetric stretching in the CN spectrum of TFA, which is characteristic of an organic solvent [63]. Also, a new band appears at 1745 cm^−1^ due to the presence of carbonyls of fluoroacetyl ester groups as a result of dissolution and chemical alteration of the cellulose with the TFA [64]. With respect to CN ILs, in the interaction of cellulose with BMIMCl, the cellulose serves as electron pair donor, and hydrogen atoms act as an electron acceptor. BMIMCl acts as electron donor–electron acceptor, where the oxygen and hydrogen atoms of hydroxyl groups of cellulose are separated by the cation and anion of BMIMCl to dissolve the cellulose [65]. Therefore, the interaction of BMIMCl-cellulose can be observed in the spectrum by decrease in the band at 3420 cm^−1^ and its shift at 3332 cm^−1^ due to the interaction of BMIM^+^-O-cellulose.

### 3.5. Thermogravimetric Analysis (TGA)

Figure 5 illustrates the TGA outcomes through weight loss percent as a function of the temperature of agave bagasse (AB), agave cellulose (AC), cellulose nanofibers with organic solvents (CN), and cellulose nanofibers with ionic liquids (CN ILs). There was weight loss in almost all samples in the first step; the higher weight loss of about 15% of moisture in the CN at 152 °C can be interpreted as a result of the high surface area of nanofibers and aqueous structure, whereas the 2.2% is attributed to the TFA in the nanofiber that do not interact with cellulose nanofibers can be considered as residual solvent. The weight loss of 3.5% in the CN ILs is attributed to the presence of water, while the weight loss of BMIMCl was less than 0.5%. This may be attributed to the high solubility of BMIMCl in the cellulose nanofiber, as shown in FTIR spectrum of CN ILs. However, agave bagasse weight loss was 6% at 163 °C because lignin has a more organic structure, decreasing the physical interaction with water; its rigidity allows that temperature to be displaced at higher values. In a second step, CN gave a decomposition temperature of 260 °C, and cellulose showed up at 302 °C. Agave bagasse gave a temperature of 324 °C attributed to the mixture of hemicellulose, cellulose, and lignin [66]. This displacement was attributed to the lignin, which has a decomposition temperature of 440 °C [67]. Decomposition of CN ILs was observed at 302 °C; they have a high initial weight loss beginning at 210 °C and ending at 320 °C, so it is demonstrated that the thermal decomposition of cellulose begins after this temperature [68,69,70]. The second decomposition peak appears in the second step in CN ILs at 360 °C. The thermal decomposition was carried out at a higher temperature range for nanofibers solubilized with BMIMCl, where cellulose nanofibers start after 210 °C. However, several reports describe the decomposition of cellulose at 300 °C [70,71,72]. Otherwise, in CN, there was a decrease in decomposition temperature. This behavior is attributed to the structure of cellulose, which has low crystallinity from interaction with the solvent, which produces swelling [42]. Wang and colleagues [73] reported thermal degradation of cellulose that has low crystallinity by TGA and demonstrated that it was degraded at lower temperatures. Therefore, we can assume that nanofibers prepared in this study exhibit this behavior due to the low or null cellulose crystallinity shown by XRD. Furthermore, the third peak that appeared in all samples after 450 °C for CN and cellulose and 600 °C for CN ILs and lignin is attributed to the presence of minerals obtained from agave cellulose [74].

There has been an increasing interest in elaborating nanofibers from natural resources in recent years. Due to this, cellulose extracted from agave waste can be a good alternative. Consequently, CN ILs can be used in numerous applications.

## 4. Conclusions

This study demonstrates that BMIMCl (ILs) is an effective solvent for cellulose, conferring important properties such as better solubility and consequently greater ease of nanofiber elaboration. ILs allowed the obtainment of nanofibers of cellulose from *Agave tequilana* Weber var. *azul* bagasse by the electrospinning technique, giving an additional application to this residue.

On the other hand, the nanofiber diameter of CN ILs (35 to 76 nm) has the advantage of having more area for contact with the fiber, which can be useful for applications in several areas such as green packaging, food, printable labels, pharmaceuticals, materials, biomedical implants, electronic devices, optical products, etc. For example, this method could be used to prepare composites coated with functional nanoparticles that could be applied in many fields, such as antimicrobial films, biomedicine, tissue engineering scaffolds, flame retardants, bioplastics, transparent films, food coating, membranes for water and wastewater treatment, veterinary applications, and drug delivery systems, among others. Finally, ILs used as a solvent provide a new and versatile platform for the wide utilization of cellulose resources such as agave bagasse with the aim to create novel functional materials and reduce the high pollution and high energy consumption of traditional techniques. In addition, ILs could increase the yield, selectivity, and solubility of cellulose due to their multiple advantages such as tunable structure, excellent dissolution ability, high thermal stability, non-flammability, low vapor pressure, high ionic conductivity, and miscibility with many other solvent systems.

## Figures and Tables

**Figure 1 nanomaterials-12-02819-f001:**
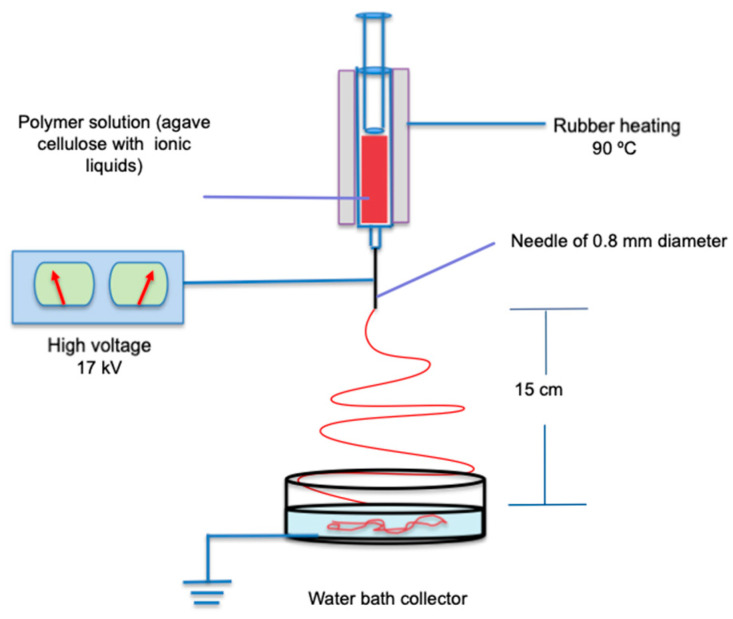
Schematic of nanofiber elaboration by electrospinning process.

**Figure 2 nanomaterials-12-02819-f002:**
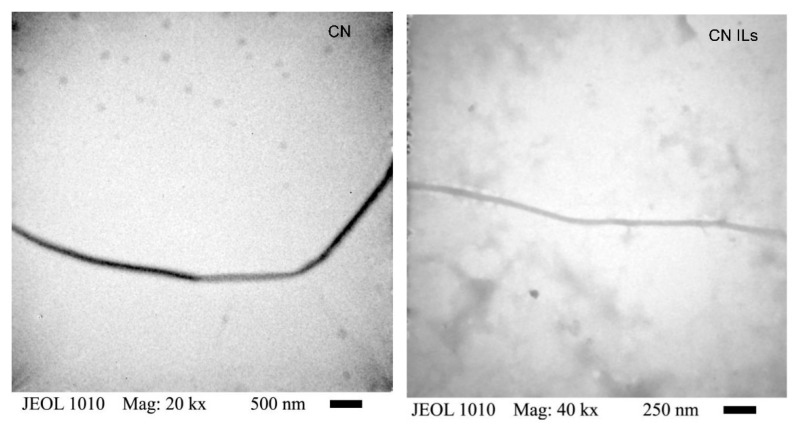
TEM micrograph of cellulose nanofiber from agave bagasse with organic solvent (CN) diameter of 88 nm, cellulose nanofiber from agave bagasse with ionic liquids (CN ILs) with diameter of 41.47 nm from electrospinning technique.

**Figure 3 nanomaterials-12-02819-f003:**
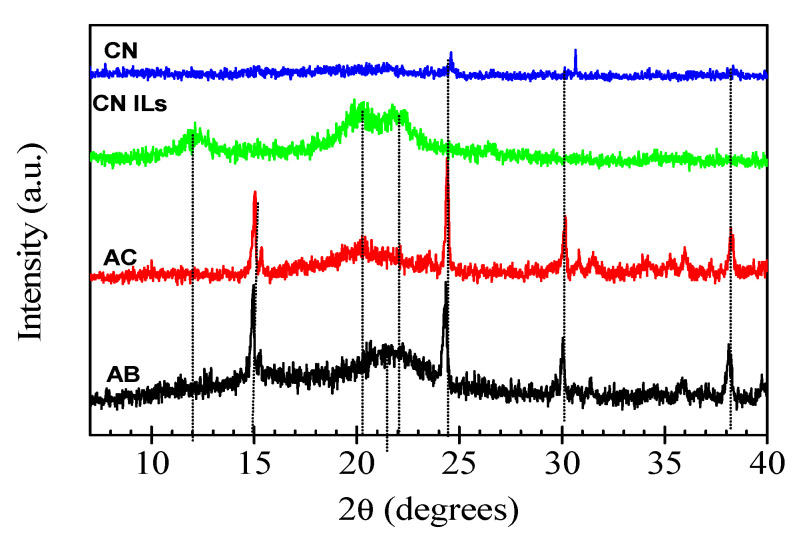
XRD diffractograms of agave bagasse (AB), agave cellulose (AC), cellulose nanofibers with ionic liquids (CN ILs), and cellulose nanofibers (CN).

**Figure 4 nanomaterials-12-02819-f004:**
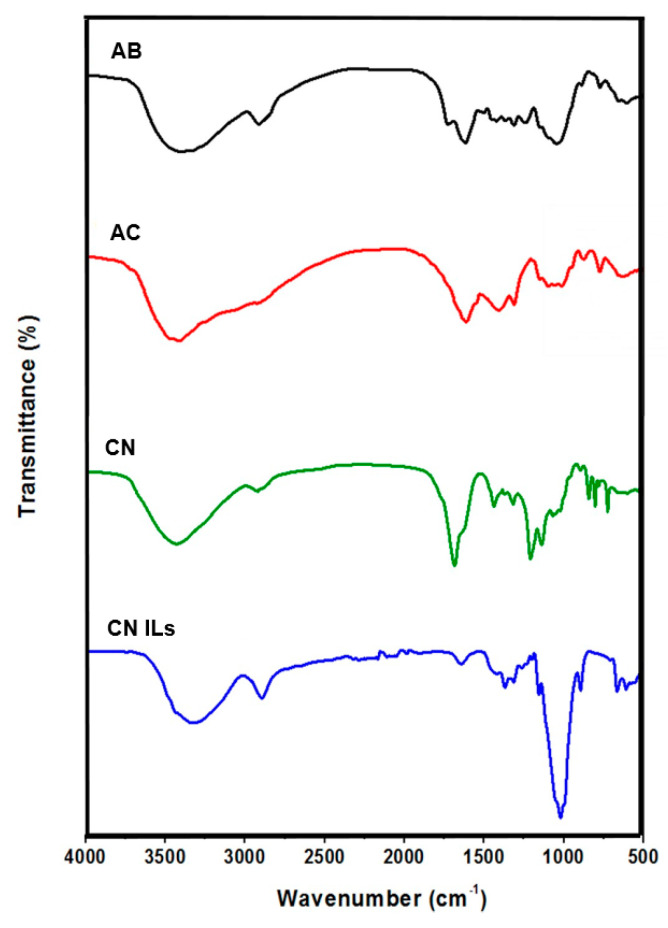
ATR-FTIR spectra of agave bagasse (AB), agave cellulose (AC), cellulose nanofibers with organic solvents (CN), and cellulose nanofibers with ionic liquids (CN ILs).

**Figure 5 nanomaterials-12-02819-f005:**
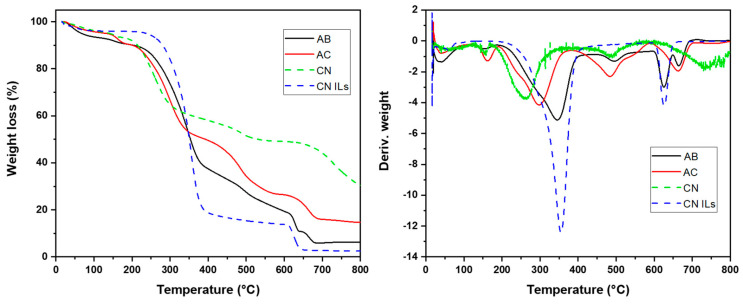
TGA of agave bagasse (AB), agave cellulose (AC), cellulose nanofibers with organic solvent (CN), and cellulose nanofibers with ionic liquid (CN ILs).

**Table 1 nanomaterials-12-02819-t001:** Polymer concentration in ionic liquid used to fabricate nanofibers from agave bagasse cellulose.

Cellulose (g)	BMIMCl (g)	Polymer Concentration (wt%)	DMSO (g)	Dissolution Temperature (°C)	Dissolution Time (min)	Result
0.015	1.5	1	0.001	90	35	Dropped
0.030	1.5	2	0.001	90	35	No jet formed
0.040	1.5	2.5	0.001	90	35	No jet formed
0.063	1.5	4	0.005	90	45	Jet formed
0.075	1.5	5	0.005	90	45	Not jet formed
0.12	1.5	8	0.005	90	45	Not jet formed

## Data Availability

The data generated from this research are available from the authors.

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
