# Peer review of "Effect of Ionic Liquids in the Elaboration of Nanofibers of Cellulose Bagasse from Agave tequilana Weber var. azul by Electrospinning Technique"

_nanomaterials, 2022, doi:10.3390/nano12162819_

Round 1
Reviewer 1 Report
This work reports on the refinement of cellulose nanofibers derived from Agave tequilana Weber var. azul by electrospinning method. Authors show that by using ionic liquids (1-butyl-3-methylimidazolium chloride, BMIMCl) as solvent, the diameter of fibers can be elaborated to 35-76 nm compared to the case of using TriFluorAcetic acid (TFA) as solvent. This work tries to prove that ionic liquids functionalized cellulose nanofiber have better crystallinity and enhanced thermal property. However, the evidence demonstrated herein is not adequate. Typical issues are listed as follows.
1. The micro structures of as-prepared nanofifer should be further detailed. At least a large area SEM and TEM image should be provided to reveal the distribution of size.
2. Authors show that cellulose fibers are crystalline, so a high resolution TEM image is needed.
3. Why is the weight loss of CN ILs so low?
4. The mechanism that the diameter of CN ILs is refined should be discussed. Compared to TFA as solvent, why does ILs lead to better spinnability?
5. I see numerous grammar mistakes in this work. For example, symbols of temperature are irregular. Even in the Figure 1, I see typos, “water bath colector”.
6. The size of Figures should be reorganized. Figure 1 and Figure 4 are over sized.
7. At least an application demonstration of as-prepared CN ILs be shown.
8. This work focused on the polymers. So I think this work suits better a specialized Journal relevant to polymers
Author Response
We appreciate the reviewer's time and comments and suggestions to increase the quality of the article.
- The micro structures of as-prepared nanofifer should be further detailed. At least a large area SEM and TEM image should be provided to reveal the distribution of size.
Response 1. When we analyzed the nanofibers we could only obtain the images that are included in the manuscript. It is worth mentioning that we do not have a new high-resolution microscope and it presented several problems when reading the samples. The best micrographs we could obtain were those presented in the manuscript. We apologize for it.
- Authors show that cellulose fibers are crystalline, so a high resolution TEM image is needed.
Response 2. When we analyzed the nanofibers we could only obtain the images that are included in the manuscript as described in response 1.
- Why is the weight loss of CN ILs so low?
Response 3. We added the explanation in the document in the subject, thermogravimetric analysis: The weight loss of 3.5% in the CN ILs is attributed to the presence of water, while the weight of BMIMCl was less than 0.5%, this may be attributed to the high solubility of BMIMCl in the cellulose nanofiber, as shown in FTIR spectrum of CN ILs.
- The mechanism that the diameter of CN ILs is refined should be discussed. Compared to TFA as solvent, why does ILs lead to better spinnability?
Response 4: The inclusion of ILs into polymer matrix induces a decrease in the fiber orientation, and also avoid of faulty coagulation. The coagulation of cellulose from ILs is a diffusion dependent process for which the mass transfer of ILs as well as the diffusion of the surface of the filaments and the coagulation agent are important (Krugly et al., 2022). These properties are not exhibited by TFA. For this reason, ILs for electrospinning of cellulose is an important parameter of the final morphology. This was written in the document.
- I see numerous grammar mistakes in this work. For example, symbols of temperature are irregular. Even in the Figure 1, I see typos, “water bath colector”.
Response 5. English language and style were revised by an expert. The symbols were changed and corrected.
- The size of Figures should be reorganized. Figure 1 and Figure 4 are over sized.
Response 6: The size of the figures were adjusted.
- At least an application demonstration of as-prepared CN ILs be shown.
Response 7. Considering the characteristic of the fiber obtained we supposed that this material has potential applications on several sector such as drug delivery systems, reinforcing materials, elaborations of antimicrobial films among others, based on reports that have similar characteristics. This was written in the document. We know that it is important to demonstrate a possible application in a practical way, however, this article is only focused on the characterization of nanofibers and another article will be published later with several tests carried out to provide the information indicated by the reviewer. Currently, it is working on that.
- This work focused on the polymers. So I think this work suits better a specialized Journal relevant to polymers
Response 8. We appreciate the reviewer's suggestion. However, we decided to submit the article to nanomaterials because the fibers obtained fall on the nanometric scale and within the scope of the journal it contemplates the characterization of nanomaterials.

Reviewer 2 Report
The authors display the bagasses from some kind of plant for preparing nanofibres to reinforce the ionic liquid. It sounds good and have some clear results. However, I think it still lacks the SEM for the general morphology demonstration. In addition, detailed discussion about the FTIR to explore the interface between nanofibres and substrate is necessary here. I think this paper can be accepted after solved those problems.
Author Response
We appreciate the reviewer's time and comments and suggestions to increase the quality of the article.
- The authors display the bagasses from some kind of plant for preparing nanofibres to reinforce the ionic liquid. It sounds good and have some clear results. However, I think it still lacks the SEM for the general morphology demonstration.
Response 1. When we analyzed the nanofibers we could only obtain the images that are included in the manuscript. It is worth mentioning that we do not have a new high-resolution microscope and it presented several problems when reading the samples. The best micrographs we could obtain were those presented in the manuscript. We apologize for it.
- In addition, detailed discussion about the FTIR to explore the interface between nanofibres and substrate is necessary here.
Response 2. We added a detailed discussion in the document in the subject, FTIR- ATR Spectroscopy: Also, a new band appears at 1745 cm-1 due to the presence of carbonyls of fluoroacetyl ester groups as a result of dissolution and chemical of the cellulose with the TFA [Hasegawa et al, 1992]. Respect to CN ILs, in the interaction of cellulose with BMIMCl, the cellulose serves as electron pair donor and hydrogen atoms act as an electron acceptor. BMIMCl acts as electron donor-electron-acceptor, where the oxygen and hydrogen atoms of hydroxyl groups of cellulose are separated by the cation and anion of BMIMCl to dissolve the cellulose [Feng and Chen, 2008]. Therefore, the interaction BMIMCl-cellulose can be observed in the spectrum by decrease in the band at 3420 cm-1, and its shift at 3332 cm-1 due to the interaction of BMIM+-O-cellulose.

Reviewer 3 Report
1. In the preparation of polymer solution, the authors used ionic liquids (BMIMCl) with small amount of DMSO to dissolve agave bagasse cellulose. However, in section of abstract and introduction, the role of DMSO on their dissolving performance was not discussed
2. With respect to cellulose yield. I am very confused why do you discuss this section? Because extraction of cellulose from agave bagasse has been reported in the following literature.
Robles-García, M. Á.; Del-Toro-Sánchez, C. L.; Márquez-Ríos, E.; Barrera-Rodríguez, A.; Aguilar, J.; Aguilar, J. A.; Rey-noso-Marín, F. J.; Ceja, I.; Dórame-Miranda, R.; Rodríguez-Félix, F. Nanofibers of cellulose bagasse from Agave tequilana Weber var. azul by electrospinning: preparation and characterization. Carbohydr. Polym. 2018, 192, 69-74.
3. Why did you compare 1-butyl-3-methylimidazolium chloride with the organic solvent TriFluorAcetic acid? Why not compare other ILs?
4. In line 245, the sentence “As far as we know, there are no reports about the production of cellulose nanofibers from agave bagasse using ILs as a solvent obtained by electrospinning. Therefore, this is an alternative to use this agro-industrial waste that could be employed in many applications in different industrial sectors.” should be deleted. Because cellulose nanofibers using various ILs obtained by electrospinning have been investigated extensively.
5. The conclusion should be improved.
Author Response
We appreciate the reviewer's time and comments and suggestions to increase the quality of the article.
- In the preparation of polymer solution, the authors used ionic liquids (BMIMCl) with small amount of DMSOto dissolve agave bagasse cellulose. However, in section of abstract and introduction, the role of DMSO on their dissolving performance was not discussed
Response 1. We added in the abstract that DMSO were added as co-solvent, and a detailed information about the function of the DMSO was described in the introduction section.
- With respect to cellulose yield. I am very confused why do you discuss this section? Because extraction of cellulose from agave bagasse has been reported in the following literature. Robles-García, M. Á.; Del-Toro-Sánchez, C. L.; Márquez-Ríos, E.; Barrera-Rodríguez, A.; Aguilar, J.; Aguilar, J. A.; Rey-noso-Marín, F. J.; Ceja, I.; Dórame-Miranda, R.; Rodríguez-Félix, F. Nanofibers of cellulose bagasse from Agave tequilana Weber var. azul by electrospinning: preparation and characterization. Polym. 2018, 192, 69-74.
Response 2. The reviewer is right. This section was removed.
- Why did you compare 1-butyl-3-methylimidazolium chloride with the organic solvent TriFluorAcetic acid?
Response 3. Cellulose is neither meltable nor soluble in conventional solvents, which limits the extent of its application, for this reason we compared 1-butyl-3-methylimidazolium chloride with the organic solvent TriFluorAcetic acid (TFA) due that TFA have negative impacts on the environment, toxicity, cost, low dissolving capacity, and difficulty in solvent recovery. On the other hand, the imidazolium cation structure derivates physical and chemical properties can be adjusted by changing the anion chains side. Also, TFA increases the jet´s charge density, and it also increases elongation forces of the jet, enabling the fibers to be assembled with smaller diameters. This does not happen with other solvents.
- Why not compare other ILs?
Response 4. There are several ILs with melting points below 100 ºC could dissolve cellulose, but the most successful ionic liquid was BMIMCl with the greatest dissolving capability of up to 10 wt% concentration of cellulose by heating. For this reason, we selected this ILs.
- In line 245, the sentence “As far as we know, there are no reports about the production of cellulose nanofibers from agave bagasse using ILs as a solvent obtained by electrospinning. Therefore, this is an alternative to use this agro-industrial waste that could be employed in many applications in different industrial sectors.” should be deleted. Because cellulose nanofibers using various ILs obtained by electrospinning have been investigated extensively.
Response 5. We removed this part as reviewer suggested.
- The conclusion should be improved.
Response 6. The conclusion was improved.

Round 2
Reviewer 1 Report
I noticed the authors have tried to improve the work although no additional experiments were conducted due to objective reasons. I recommend accept of this work for Nanomaterials.
Reviewer 3 Report
this manuscript can be accepted in present form